# Impact of Gypsum and Bio-Priming of Maize Grains on Soil Properties, Physiological Attributes and Yield under Saline–Sodic Soil Conditions

Megahed M. Amer [1], Mohssen Elbagory [2,3], Sahar El-Nahrawy [3] and Alaa El-Dein Omara [3,*]

[1] Agricultural Research Center, Department of Soil Improvement and Conservation, Soils, Water and Environment Research Institute, Giza 12112, Egypt
[2] Department of Biology, Faculty of Science and Arts, King Khalid University, Mohail 61321, Assir, Saudi Arabia
[3] Agricultural Research Center, Department of Agricultural Microbiology, Soils, Water and Environment Research Institute, Giza 12112, Egypt
* Correspondence: alaa.ahmed@arc.sci.eg or alaa.omara@yahoo.com

**Abstract:** In order to enhance soil qualities and boost crop output, gypsum, plant-growth-promoting rhizobacteria (PGPR), and chitosan are all viable solutions. This study's goal was to find out how different amounts of chitosan—0, 25, 50, 75, and 100 mg L$^{-1}$—in combination with gypsum and PGPR inoculation (*Azospirillum lipoferum* + *Bacillus subtilis*), affected the yield of maize plants growing in saline–sodic soil. Field tests were carried out in triplicate across two growing seasons, 2020 and 2021, using a split plot design. According to the findings, applying the gypsum + PGPR + 50 mg L$^{-1}$ chitosan treatment (T8) considerably improved plant physiology (chlorophyll, carotenoids, and proline levels), nutrient indicators (N, K$^+$ and K$^+$/Na$^+$ ratio), soil enzyme activity (dehydrogenase, urease, amylase, and invertase), cation exchange capacity (CEC), and porosity. On the other hand, we revealed positive effects on Na$^+$, bulk density (BD), electrical conductivity (EC), and the proportion of exchangeable sodium (ESP), thus, enhancing the productivity compared to the alternative treatment. Therefore, it might be inferred that using gypsum, microbial inoculation, and 50 mg L$^{-1}$ chitosan may be a key strategy for reducing the detrimental effects of salinity on maize plants.

**Keywords:** gypsum; chitosan; microbial inoculation; soil physicochemical properties; productivity of maize

## 1. Introduction

A severe environmental stressor and barrier to crop production in dry and semi-arid regions, yields are reduced by 25 to 30% by salinity stress [1,2]. Salinization of arable land is predicted to have severe repercussions on the entire world, losing up to 50% of land by the middle of the twenty-first century, and 30% of land during the following 25 years [3], and there is currently a loss of roughly USD 12 billion due to salt stress on 50% of the world's agricultural land [4].

Globally, maize is an economically important cereal crop whose productivity is affected by high soil salinity. In Egypt, harvested area has reached about 1,458,881 ha for that production quantity of 7.5 million tons of grains [5]. The reason of crop productivity reduction is due to the physiological and biochemical processes in the plant, which are negatively affected by the complex process of soil salinity [6]. Therefore, the toxicity of ions during osmotic stress leads to a decrease in plant growth and the reversal of nutrient absorption and enzyme activity [7,8]. On the other hand, salt-affected soils have negative effects that include reduced osmosis, cytotoxicity of ions, generation of ROS (reactive oxygen species), and nutrient imbalance, which causes water stress [9,10]. The effects of salinity stress on the growth dynamics of different crops depend on the salt concentration, time period, plant species, and gas exchange capabilities, as well as environmental conditions [11].

Therefore, it is crucial to investigate sustainable management methods to lessen this issue's negative effects through genetic engineering, crop breeding for salinity tolerance, soil conditioners, and biological agents [12–14]. Enhancing food crops' ability to continue growing and producing under soil salinity stress conditions has been the focus of several studies on salinity management. The majority of these initiatives aim to keep ideal $K^+/Na^+$ ratios, nutrient ratios in soil solution, organic matter, as well as improved antioxidant and amino acid synthesis in plants, at optimal levels [15,16]. In this regard, gypsum has been shown to support ideal $K^+/Na^+$ and $Ca^{2+}/Na^+$ ratios, lowered pH, as well as providing crops with the necessary S nutrition in salty soils on multiple studies [17,18], and it has also been shown to facilitate the synthesis of chlorophyll, proteins, and the increased uptake of P and N [19]. On the other hand, the ability of beneficial microorganisms (*Azospirillum*, *Bacillus*, *Pseudomonas* and *Rhizobium*, etc.) to transformation nutrients from organic and inorganic into soluble forms, plays a significant role in reducing salt stress and appears to be a promising process for increasing crops' tolerance to salinity [20].

Some organic chemicals are also utilized to reduce salt stress on plants, in addition to soil conditioners (gypsum) and beneficial microorganisms. One of these materials generated from chitin is chitosan (polysaccharide biopolymer), which is produced from krill and shrimp [21]. Chitosan is a polysaccharide that is biodegradable, renewable, non-toxic, and biocompatible. It is also categorized as an elicitor in plants, activating genes that are involved in pathways for the secondary metabolite biosynthesis [22,23], and the effectiveness of the application of chitosan depends on the concentration, water content, temperature, and growth stage of the plant. Chitosan can be applied to plant organs to encourage the accumulation of bioactive secondary metabolites both in vivo and in vitro [23], decrease plant transpiration in pepper plants, which previously led to a 26–43% decrease in water use without a change in dry matter production [24], and improve chlorophyll content and root establishment in plants [25].

From the above-mentioned studies, gypsum has a significant potential for improving saline soils when coupled with beneficial bacteria and chitosan, which boost the organic carbon, humus, nutrient contents of the soil and improve the drop production. The current study postulated that applying gypsum, PGPR, and various concentrations of chitosan during the 2020 and 2021 growing seasons would ultimately improve plant physiology, nutrient contents, soil enzyme activity, soil physicochemical properties, and the productivity of maize plants grown in saline–sodic soil.

## 2. Materials and Methods

### 2.1. Description of the Location

This experiment was carried out at the research farm of SARS (Latitude: 31°6′ N and Longitude: 30°56′ E), Kafr El-Sheikh, which is situated in the northern part of Egypt, along the western branch of the Nile in the Nile Delta. The meteorological data for the experiment location are shown in Table 1.

**Table 1.** Meteorological data for the two summer growing seasons of 2020 and 2021.

| Month | Temperature (°C) | | | | Wind Speed (km day$^{-1}$) | | RH (%) | | Rainfall (mm month$^{-1}$) | |
|---|---|---|---|---|---|---|---|---|---|---|
| | Max | Min | Max | Min | | | | | | |
| | 2020 | | 2021 | | 2020 | 2021 | 2020 | 2021 | 2020 | 2021 |
| May | 33.2 | 17.1 | 33.5 | 17.4 | 125.1 | 124.8 | 68.3 | 69.8 | 0.0 | 0.0 |
| June | 36.4 | 16.5 | 36.5 | 16.8 | 115.1 | 113.7 | 65.2 | 67.7 | 0.0 | 0.0 |
| July | 35.6 | 22.0 | 35.7 | 22.3 | 105.4 | 102.5 | 64.3 | 66.2 | 0.0 | 0.0 |
| August | 38.4 | 23.3 | 38.1 | 23.6 | 93.1 | 91.3 | 62.1 | 64.4 | 0.0 | 0.0 |
| September | 37.2 | 22.2 | 37.5 | 22.7 | 85.0 | 86.2 | 45.6 | 48.5 | 0.0 | 0.0 |

Max—maximum, min—minimum, RH—relative humidity.

### 2.2. Materials Used and Their Source

Gypsum, plant-growth-promoting rhizobacteria (PGPR) and chitosan, were applied as soil amendments to the grains treated. Gypsum was supplied from a soil improvement



device, Sakha Agricultural Research Station (SARS), Kafr El-Sheikh, Egypt, which was used according to the following equation:

$$GR = (ESP_i - ESP_f)/100 \times CEC \times 1.72 \times (100/purity)$$

where, GR: gypsum requirement (Mg ha$^{-1}$), ESP$_i$: initial soil ESP, ESP$_f$: The desired soil ESP, and CEC: cation exchange capacity (cmol$_c$ kg$^{-1}$); 1.72 tons is the amount of CaSO$_4 \cdot$2H$_2$O required to reduce the Na$^+$ content of the soil [26]. For PGPR, two strains of *Azospirillum lipoferum* SP2 and *Bacillus subtilis* MF497446 were provided by the Department of Agricultural Microbiology, SWERI, ARC, Egypt. *A. lipoferum* was grown on a semi-solid malate medium [27], and *B. subtilis* was grown on nutrient broth medium [28]. From Sigma–Aldrich, chitosan (CAS Number: 9012-76-4) was purchased and dissolved in 1% acetic acid solution according to [29].

*2.3. Field Experiments and Growth Conditions*

Two field experiments (summer 2020 and 2021) were conducted in a split- plot design with three replicates to study the effect of different levels of chitosan, inoculation with PGPR and gypsum applications on some physical and chemical properties of soil, soil enzymes, physiological traits, and the yield of maize (*Zea mays* L. cv. Hybrid 10) in saline–sodic soil. Soil properties (physicochemical and biological) of the experimental location for the two summer growing seasons are presented in Table 2. The main plots were divided into two treatments of gypsum (without gypsum and with gypsum), and the sub-plots were T1: without PGPR + 0 mg L$^{-1}$ chitosan; T2: without PGPR + 25 mg L$^{-1}$ chitosan; T3: without PGPR + 50 mg L$^{-1}$ chitosan; T4: without PGPR + 75 mg L$^{-1}$ chitosan; T5: without PGPR + 100 mg L$^{-1}$ chitosan; T6: with PGPR + 0 mg L$^{-1}$ chitosan; T7: with PGPR + 25 mg L$^{-1}$ chitosan; T8: with PGPR + 50 mg L$^{-1}$ chitosan; T9: with PGPR + 75 mg L$^{-1}$ chitosan; T10: with PGPR + 100 mg L$^{-1}$ chitosan.

**Table 2.** Average values of some physicochemical and biological properties of the experimental site during the two cropping seasons.

| Depth (cm) | Soil Physical Properties | | | | | | | |
|---|---|---|---|---|---|---|---|---|
| | Moisture Characteristics | | | | Particle Size Distribution (g kg$^{-1}$) | | | |
| | F.C (%) | W.P. (%) | A.W. (%) | B.D. (kg m$^{-3}$) | Sand | Silt | Clay | Soil Texture |
| 0–20 | 44.11 | 22.01 | 22.10 | 1.29 | 173.1 | 255.1 | 571.8 | clay |
| 20–40 | 40.52 | 20.28 | 20.24 | 1.31 | 188.5 | 247.6 | 563.9 | clay |
| 40–60 | 38.03 | 19.03 | 19.00 | 1.33 | 190.6 | 251.2 | 558.2 | clay |

| Depth (cm) | Soil Chemical Properties | | | | | |
|---|---|---|---|---|---|---|
| | pH | EC (dSm$^{-1}$) | ESP (%) | CEC (cmole kg$^{-1}$) | OM (g kg$^{-1}$) | CaCO$_3$ (g kg$^{-1}$) |
| 0–20 | 8.17 | 6.70 | 14.75 | 37.15 | 17.8 | 27.8 |
| 20–40 | 8.20 | 7.61 | 16.90 | 36.10 | 16.4 | 26.9 |
| 40–60 | 8.35 | 10.89 | 18.98 | 34.42 | 14.2 | 24.1 |

| Soil depth (cm) | Biological Properties (cfu g$^{-1}$ dry weight soil) | | |
|---|---|---|---|
| | TCB | TCF | TCA |
| 0–20 | 123 × 10$^6$ | 71 × 10$^4$ | 67 × 10$^5$ |

**F.C.**: Field Capacity; **W.P.**: Wilting Point; **A.W.**: Available Water; **B.D.**: Bulk Density; **pH**: determined in soil water suspension (1:2.5); **EC**: determined in saturated soil paste extract; **ESP**: Exchangeable Sodium Percent; **CEC**: Cation Exchange Capacity; **OM**: Organic Matter; **TCB**: Total count of bacteria; **TCF**: Total count of fungi; **TCA**: Total count of actinomycetes.

The experimental unit (sub-plots) was made up of five ridges, each measuring 4 m in length and 60 cm apart and the grains were planted at a rate of 2:3 grains hole$^{-1}$ with 25 cm spacing in between, and the distance between replications was 100 cm. The Maize Research Department, SARS, Egypt, supplied the grains of maize which were sown on 28 May 2020, and 2 June 2021, at a rate of 30 kg ha$^{-1}$.

Before sowing, grains were soaked for 12 h in chitosan solutions containing varying concentrations [30], after which they were combined with PGPR (950 g h$^{-1}$, 237.5 mL of $10^8$ CFU mL$^{-1}$ from each culture per 475 g of the sterilized carrier and mixed carefully with maize grains using a sticking material) and thinned for one seedling hole$^{-1}$ after germination. In addition to applying soil amendment, gypsum was incorporated into the soil surface (0–30 cm, equivalent to 9.73 ton ha$^{-1}$) with the plowing processes. For mineral fertilizer, P and K fertilizers were applied at a rate of 480 and 120 kg ha$^{-1}$ as calcium superphosphate and potassium sulfate, during the seedbed preparation, respectively. At 24 and 36 days from sowing, N fertilizer was applied at a rate of 290 Kg ha$^{-1}$ as urea. According to the recommendation of the Ministry of Agriculture, the plants were irrigated every 12 days, and agricultural methods were applied during the cropping seasons.

*2.4. Biometrics of Plants*

2.4.1. Photosynthetic Pigments

Chlorophyll (Chl) and carotenoids (Car) are photosynthetic pigments that absorb solar energy for photosynthesis and are susceptible to various environmental conditions. The amount of total chlorophyll and carotenoids in tissues taken from the second completely developed leaf at the plant's tip were measured sixty days after the grains were sown. According to [31], the photosynthetic pigments' content was calculated. Concisely, 0.1 g of fresh leaf tissue was ground in 5 mL of acetone 80%, followed by a 10-min centrifugation process at 13,000 rpm. Using a UV spectrophotometer (Model 6705), the supernatant's absorbance was measured at 645, 663, and 470 nm, to determine the extract's chlorophyll (mg g$^{-1}$ FW) and carotenoid ($\mu$g g$^{-1}$ FW) contents.

2.4.2. Proline Content

After 60 days from the date of sowing [32], we measured the endogenous proline content in the second completely grown leaf from the plant tip. Concisely, 0.1 g of fresh plant tissues was completely combined with 4 mL of 3.0% sulfosalicylic acid in a mortar and stored at 5 °C overnight. The suspension was centrifuged at room temperature for 5 min at 3000 rpm. With the supernatant, 4 milliliters of acidic ninhydrin reagent was mixed. After being mechanically shaken, tubes were heated in a bath of boiling water for one hour. The mixture was then extracted with 4 mL of toluene in a separating funnel once the tubes were cooled. By using spectrophotometry, the absorbance of the toluene layer was measured at 520 nm. With reference to the standard curve, the concentration of the unidentified sample was estimated. Six samples on average were used for each treatment in the final value.

2.4.3. Determination of N, K$^+$, Na$^+$ and the K$^+$/Na$^+$ Ratio

Six leaves (the second fully-grown leaf) from each treatment were sampled 60 days after the grains were sown and dried in an oven at 70 °C for 2 days. A fine powder made from 0.5 g of dried leaves was then added to Kjeldahl digestion tubes together with 5 mL of sulfuric acid (95–97%, Merck). Once the tubes were on the heater, the temperature was raised gradually (5 °C/min) until it reached 270 °C, where digestion continued for 2 h. After cooling for 30 min, 1 mL of perchloric acid (80%, Merck) was added to the samples. The temperature was then raised to 150 °C for another 1 h, until the digesting solution became clear. The sample was diluted to a level of 50 mL in a volumetric flask using ultra-pure water. According to [33], nitrogen percentage was measured by micro-Kjeldahl's and using a flame photometer; the K$^+$, Na$^+$, and K$^+$/Na$^+$ ratio were measured using the techniques of [34].

### 2.4.4. Soil Enzymes Estimations

Triphenyl formazon (TPF, red-colored) was created by reducing 2, 3, 5- triphenylote-trazolium chloride (TTC) in order to measure the dehydrogenase activity (mg TPF g$^{-1}$ soil day$^{-1}$) of soil samples according to [35]. Additionally, the amount of ammonium created by urea hydrolysis in soil was assessed as the amount of urease enzyme activity (mg NH$_4^+$-N g$^{-1}$ soil day$^{-1}$) in soil samples [36]. We used starch as a substrate, which measures the amylase activity (mg glucose g$^{-1}$ soil h$^{-1}$) of various soil treatments as indicated by [37]. In addition, using sucrose as a substrate, invertase activity (mg sucrose g$^{-1}$ soil h$^{-1}$) of soil samples was assessed in accordance with [38].

### 2.4.5. Soil Physicochemical Characteristics

Soil samples (30 cm depth) were gathered and homogenized as a single sample per replicate at 120 days after sowing (harvest). The soil samples were crushed and put through a 2 mm filter after being air-dried. An EC meter was used to measure the electrical conductivity (EC, dS m$^{-1}$) in the soil paste extract (Genway, Dunmow Essex, UK). According to [39], measurements were made of the cation exchange capacity (CEC, cmole kg$^{-1}$) and exchangeable sodium percentage (ESP, %). On the other hand, soil bulk density (BD, kg m$^{-3}$) and total porosity (TP, %) were determined using the core sampling technique as described by [40].

### 2.5. Yield

Harvesting was carried out four months after seeding at a moisture level of 15.5%, and ten plants were randomly selected from the fourth inner ridges to calculate yield including the length and diameter of ear (cm), 100-grain weight, and the grain yield (kg ha$^{-1}$).

### 2.6. Statistical Analysis

Co Stat's statistical package software, version 6.303, was used to statistically evaluate the data. ANOVA was used to compare the various treatments. Tukey's range tests were used for multiple comparisons at $p \leq 0.05$ [41].

### 3. Results

### 3.1. Photosynthetic Pigments

At 60 days after sowing, maize leaves from both the 2020 and 2021 cropping seasons contained significantly different amounts ($p \leq 0.05$) of chlorophyll, carotenoids, and proline depending on the applications of gypsum (without gypsum and with gypsum), combinations of PGPR and different levels of chitosan (0, 25, 50, 75, and 100 mg L$^{-1}$ (Table 3)).

With application of the gypsum + PGPR + 50 mg L$^{-1}$ chitosan treatment (T8), we obtained 2.45 and 0.85 in 2020 season and 2.54 and 0.94 in 2021 season, respectively; extremely significant differences were seen for the chlorophyll and carotenoids parameters. The same pattern was seen for proline content as well. For instance, the highest values for T8 (50 mg L$^{-1}$), T9 (75 mg L$^{-1}$), and T10 (100 mg L$^{-1}$), treatments, were 10.53, 10.48, and 10.47 μmole g$^{-1}$ FW in season 2020 and 10.62, 10.57, and 10.56 μmole g$^{-1}$ FW in season 2021, respectively (Table 3). Consequently, at various concentrations of chitosan under gypsum + PGPR treatments, the results were in the descending order of 50 > 75 > 100 > 25 > 0 mg L$^{-1}$.

**Table 3.** Combined effects of gypsum, and bio-priming with PGPR and different levels of chitosan (0, 25, 50, 75, and 100 mg $L^{-1}$), on chlorophyll, carotenoid and proline contents of leaves in maize plants at 60 days from sowing during 2020 and 2021 seasons.

| Treatments | | Chlorophyll (mg $g^{-1}$ FW) | | Carotenoid (µg $g^{-1}$ FW) | | Proline (µmol $g^{-1}$ FW) | |
|---|---|---|---|---|---|---|---|
| | | 2020 | 2021 | 2020 | 2021 | 2020 | 2021 |
| **Without Gypsum** | T1 | 1.16 ± 0.11 [o] | 1.09 ± 0.08 [o] | 0.44 ± 0.03 [k] | 0.42 ± 0.08 [n] | 8.24 ± 0.28 [o] | 8.35 ± 0.48 [P] |
| | T2 | 1.31 ± 0.14 [m] | 1.24 ± 0.11 [n] | 0.50 ± 0.04 [j] | 0.48 ± 0.10 [m] | 8.39 ± 0.49 [n] | 8.50 ± 0.67 [o] |
| | T3 | 1.79 ± 0.08 [i] | 1.72 ± 0.06 [i] | 0.69 ± 0.08 [e] | 0.67 ± 0.06 [g] | 8.87 ± 0.39 [k] | 8.98 ± 0.58 [l] |
| | T4 | 1.45 ± 0.05 [k] | 1.38 ± 0.13 [l] | 0.56 ± 0.05 [h] | 0.54 ± 0.03 [k] | 8.55 ± 0.41 [l] | 8.66 ± 0.68 [m] |
| | T5 | 1.40 ± 0.12 [l] | 1.33 ± 0.01 [m] | 0.54 ± 0.07 [i] | 0.52 ± 0.08 [l] | 8.48 ± 0.53 [m] | 8.59 ± 0.69 [n] |
| | T6 | 1.26 ± 0.08 [n] | 1.34 ± 0.11 [lm] | 0.45 ± 0.08 [k] | 0.51 ± 0.10 [l] | 8.97 ± 0.48 [j] | 9.05 ± 0.57 [k] |
| | T7 | 1.44 ± 0.10 [k] | 1.52 ± 0.07 [k] | 0.52 ± 0.10 [j] | 0.58 ± 0.07 [j] | 9.15 ± 0.38 [i] | 9.23 ± 0.59 [j] |
| | T8 | 2.12 ± 0.16 [d] | 2.20 ± 0.13 [de] | 0.76 ± 0.06 [c] | 0.82 ± 0.03 [c] | 9.83 ± 0.36 [fg] | 9.91 ± 0.49 [h] |
| | T9 | 1.54 ± 0.18 [j] | 1.62 ± 0.17 [j] | 0.55 ± 0.08 [hi] | 0.61 ± 0.07 [i] | 9.25 ± 0.58 [h] | 9.33 ± 0.33 [i] |
| | T10 | 1.54 ± 0.19 [j] | 1.62 ± 0.11 [j] | 0.55 ± 0.09 [hi] | 0.61 ± 0.10 [i] | 9.25 ± 0.39 [h] | 9.33 ± 0.61 [i] |
| **With Gypsum** | T1 | 1.85 ± 0.09 [h] | 1.96 ± 0.22 [h] | 0.60 ± 0.09 [g] | 0.64 ± 0.06 [h] | 9.81 ± 0.58 [g] | 9.94 ± 0.58 [h] |
| | T2 | 1.90 ± 0.08 [g] | 2.01 ± 0.19 [g] | 0.61 ± 0.08 [g] | 0.65 ± 0.09 [gh] | 9.86 ± 0.47 [f] | 9.99 ± 0.69 [g] |
| | T3 | 2.26 ± 0.05 [c] | 2.37 ± 0.09 [c] | 0.73 ± 0.05 [d] | 0.77 ± 0.09 [d] | 10.22 ± 0.58 [c] | 10.35 ± 0.69 [c] |
| | T4 | 2.05 ± 0.11 [e] | 2.16 ± 0.06 [e] | 0.66 ± 0.03 [f] | 0.70 ± 0.06 [f] | 10.01 ± 0.68 [e] | 10.14 ± 0.71 [f] |
| | T5 | 2.12 ± 0.14 [d] | 2.23 ± 0.09 [d] | 0.68 ± 0.04 [e] | 0.72 ± 0.09 [e] | 10.08 ± 0.49 [d] | 10.21 ± 0.53 [e] |
| | T6 | 1.99 ± 0.18 [f] | 2.08 ± 0.11 [f] | 0.69 ± 0.08 [e] | 0.78 ± 0.05 [d] | 10.07 ± 0.27 [d] | 10.16 ± 0.58 [f] |
| | T7 | 2.11 ± 0.19 [d] | 2.20 ± 0.12 [de] | 0.73 ± 0.09 [d] | 0.82 ± 0.06 [c] | 10.19 ± 0.49 [c] | 10.28 ± 0.61 [d] |
| | T8 | 2.45 ± 0.11 [a] | 2.54 ± 0.18 [a] | 0.85 ± 0.07 [a] | 0.94 ± 0.08 [a] | 10.53 ± 0.39 [a] | 10.62 ± 0.66 [a] |
| | T9 | 2.40 ± 0.19 [b] | 2.49 ± 0.13 [b] | 0.83 ± 0.09 [b] | 0.92 ± 0.03 [b] | 10.48 ± 0.33 [b] | 10.57 ± 0.38 [b] |
| | T10 | 2.39 ± 0.21 [b] | 2.48 ± 0.17 [b] | 0.82 ± 0.05 [b] | 0.91 ± 0.07 [b] | 10.47 ± 0.61 [b] | 10.56 ± 0.59 [b] |
| **F-test** | | | | | | | |
| **Main** | | ** | ** | ** | ** | ** | ** |
| **Sub main** | | ** | ** | ** | ** | ** | ** |
| **Interaction** | | ** | ** | ** | ** | ** | ** |

According to the Duncan's test, means denoted by various letters show significant differences between treatments ($p \leq 0.05$). Values are the means standard deviations (SD) of three replicates. **T1**: without PGPR + 0 mg $L^{-1}$ chitosan; **T2**: without PGPR + 25 mg $L^{-1}$ chitosan; **T3**: without PGPR + 50 mg $L^{-1}$ chitosan; **T4**: without PGPR + 75 mg $L^{-1}$ chitosan; **T5**: without PGPR + 100 mg $L^{-1}$ chitosan; **T6**: with PGPR + 0 mg $L^{-1}$ chitosan; **T7**: with PGPR + 25 mg $L^{-1}$ chitosan; **T8**: with PGPR + 50 mg $L^{-1}$ chitosan; **T9**: with PGPR + 75 mg $L^{-1}$ chitosan; **T10**: with PGPR + 100 mg $L^{-1}$ chitosan; **\*\***: High significant.

### 3.2. N, $K^+$, $Na^+$ and the $K^+/Na^+$ Ratio

Significant variations were seen 60 days after sowing in the percentages of N, $K^+$, and the $K^+/Na^+$ ratio, as well as $Na^+$%, in maize leaves under gypsum treatments, caused by PGPR and chitosan (Table 4). In comparison to the control treatment (T1, gypsum + PGPR + 0 mg $L^{-1}$ chitosan), soil amendments with gypsum + microbial inoculation with PGPR (*A. lipoferum* and *B. subtilis*) + 50 mg $L^{-1}$ chitosan (T8) provided the highest percentages of N and K, achieving increased rates of 25.9 and 25.2% for N and 24.08 and 22.00% for K during both 2020 and 2021, respectively (Table 4).

**Table 4.** Combined effects of gypsum, and bio-priming with PGPR and different levels of chitosan (0, 25, 50, 75, and 100 mg L$^{-1}$), on N, K$^+$, Na$^+$ and the K$^+$/Na$^+$ ratio of leaves in maize plants at 60 days from sowing during the 2020 and 2021 seasons.

| Treatments | | Nitrogen (%) | | Potassium (%) | | Sodium (%) | | Potassium/Sodium Ratio (%) | |
|---|---|---|---|---|---|---|---|---|---|
| | | 2020 | 2021 | 2020 | 2021 | 2020 | 2021 | 2020 | 2021 |
| Without Gypsum | T1 | 1.04 ± 0.11 [k] | 1.06 ± 0.03 [l] | 1.11 ± 0.09 [n] | 1.23 ± 0.04 [n] | 2.40 ± 0.14 [a] | 2.58 ± 0.11 [a] | 0.46 ± 0.03 [o] | 0.47 ± 0.05 [m] |
| | T2 | 1.19 ± 0.12 [j] | 1.21 ± 0.04 [k] | 1.26 ± 0.14 [l] | 1.38 ± 0.02 [m] | 2.29 ± 0.12 [cd] | 2.47 ± 0.09 [b] | 0.55 ± 0.02 [m] | 0.56 ± 0.06 [l] |
| | T3 | 1.67 ± 0.09 [g] | 1.69 ± 0.06 [h] | 1.74 ± 0.12 [h] | 1.86 ± 0.01 [i] | 2.09 ± 0.12 [gh] | 2.27 ± 0.13 [f] | 0.83 ± 0.04 [gh] | 0.82 ± 0.02 [h] |
| | T4 | 1.33 ± 0.04 [h] | 1.35 ± 0.03 [i] | 1.40 ± 0.10 [j] | 1.52 ± 0.06 [k] | 2.12 ± 0.09 [g] | 2.30 ± 0.15 [ef] | 0.66 ± 0.04 [k] | 0.66 ± 0.04 [k] |
| | T5 | 1.28 ± 0.07 [i] | 1.30 ± 0.02 [j] | 1.35 ± 0.08 [k] | 1.47 ± 0.05 [l] | 2.10 ± 0.16 [gh] | 2.28 ± 0.08 [f] | 0.64 ± 0.05 [kl] | 0.64 ± 0.03 [k] |
| | T6 | 1.07 ± 0.11 [k] | 1.10 ± 0.04 [l] | 1.19 ± 0.05 [m] | 1.34 ± 0.04 [m] | 2.32 ± 0.11 [c] | 2.47 ± 0.07 [b] | 0.51 ± 0.03 [n] | 0.54 ± 0.02 [l] |
| | T7 | 1.25 ± 0.15 [i] | 1.28 ± 0.03 [j] | 1.37 ± 0.07 [jk] | 1.52 ± 0.03 [k] | 2.19 ± 0.12 [f] | 2.34 ± 0.13 [d] | 0.62 ± 0.02 [l] | 0.65 ± 0.05 [k] |
| | T8 | 1.93 ± 0.10 [e] | 1.96 ± 0.03 [e] | 2.05 ± 0.07 [e] | 2.20 ± 0.06 [de] | 2.00 ± 0.15 [j] | 2.15 ± 0.12 [h] | 1.02 ± 0.01 [e] | 1.02 ± 0.06 [f] |
| | T9 | 1.35 ± 0.11 [h] | 1.38 ± 0.07 [i] | 1.49 ± 0.11 [i] | 1.62 ± 0.07 [j] | 2.03 ± 0.14 [ij] | 2.18 ± 0.11 [gh] | 0.73 ± 0.01 [j] | 0.74 ± 0.07 [j] |
| | T10 | 1.35 ± 0.09 [h] | 1.37 ± 0.02 [i] | 1.47 ± 0.15 [i] | 1.61 ± 0.05 [j] | 2.05 ± 0.08 [i] | 2.20 ± 0.14 [g] | 0.71 ± 0.02 [j] | 0.75 ± 0.07 [j] |
| With Gypsum | T1 | 1.71 ± 0.04 [g] | 1.74 ± 0.05 [g] | 1.82 ± 0.13 [f] | 1.96 ± 0.06 [h] | 2.35 ± 0.05 [b] | 2.47 ± 0.10 [b] | 0.77 ± 0.04 [i] | 0.79 ± 0.05 [i] |
| | T2 | 1.76 ± 0.03 [f] | 1.79 ± 0.03 [f] | 1.87 ± 0.09 [f] | 2.01 ± 0.03 [g] | 2.26 ± 0.11 [de] | 2.38 ± 0.12 [c] | 0.82 ± 0.05 [h] | 0.84 ± 0.04 [h] |
| | T3 | 2.12 ± 0.06 [c] | 2.15 ± 0.04 [c] | 2.23 ± 0.05 [c] | 2.37 ± 0.04 [c] | 1.92 ± 0.09 [k] | 2.04 ± 0.12 [i] | 1.16 ± 0.07 [c] | 1.16 ± 0.01 [c] |
| | T4 | 1.91 ± 0.13 [e] | 1.94 ± 0.07 [e] | 2.02 ± 0.04 [e] | 2.16 ± 0.07 [e] | 1.94 ± 0.12 [k] | 2.06 ± 0.09 [i] | 1.04 ± 0.05 [e] | 1.05 ± 0.07 [e] |
| | T5 | 1.98 ± 0.10 [d] | 2.01 ± 0.03 [d] | 2.09 ± 0.04 [d] | 2.23 ± 0.05 [d] | 1.95 ± 0.15 [k] | 2.07 ± 0.08 [i] | 1.07 ± 0.01 [d] | 1.08 ± 0.05 [d] |
| | T6 | 1.77 ± 0.08 [f] | 1.82 ± 0.02 [f] | 1.91 ± 0.07 [f] | 2.09 ± 0.03 [f] | 2.23 ± 0.16 [e] | 2.34 ± 0.09 [de] | 0.85 ± 0.04 [g] | 0.89 ± 0.06 [g] |
| | T7 | 1.89 ± 0.11 [e] | 1.94 ± 0.01 [e] | 2.03 ± 0.08 [e] | 2.21 ± 0.03 [de] | 2.09 ± 0.19 [h] | 2.20 ± 0.12 [g] | 0.97 ± 0.03 [f] | 1.00 ± 0.06 [f] |
| | T8 | 2.23 ± 0.05 [a] | 2.28 ± 0.03 [a] | 2.37 ± 0.09 [a] | 2.55 ± 0.07 [a] | 1.94 ± 0.16 [k] | 2.04 ± 0.14 [i] | 1.22 ± 0.02 [a] | 1.25 ± 0.03 [a] |
| | T9 | 2.18 ± 0.12 [b] | 2.23 ± 0.05 [b] | 2.32 ± 0.16 [b] | 2.50 ± 0.06 [b] | 1.95 ± 0.17 [k] | 2.07 ± 0.15 [i] | 1.18 ± 0.02 [b] | 1.20 ± 0.07 [ab] |
| | T10 | 2.17 ± 0.14 [b] | 2.22 ± 0.02 [b] | 2.31 ± 0.12 [b] | 2.49 ± 0.07 [b] | 1.95 ± 0.19 [k] | 2.06 ± 0.16 [i] | 1.18 ± 0.04 [b] | 1.20 ± 0.07 [b] |
| **F-test** | | | | | | | | | |
| **Main** | | ** | ** | ** | ** | ** | ** | ** | ** |
| **Sub main** | | ** | ** | ** | ** | ** | ** | ** | ** |
| **Interaction** | | ** | ** | ** | ** | ** | ** | ** | ** |

According to the Duncan's test, means denoted by various letters show significant differences between treatments ($p \leq 0.05$). Values are the means and standard deviations (SD) of three replicates. **T1–T10**: as shown in Table 3; **: High significant.

On the contrary, the maximum reduction in the Na$^+$% was in maize leaves, which decreased from 2.23% (gypsum + PGPR + 0 mg L$^{-1}$ chitosan, T6) to 2.09% (gypsum + PGPR + 25 mg L$^{-1}$ chitosan, T7), 1.94% (gypsum + PGPR + 50 mg L$^{-1}$ chitosan, T8), 1.95% (gypsum + PGPR + 75 mg L$^{-1}$ chitosan, T9), and 1.95% (gypsum + PGPR + 100 mg L$^{-1}$ chitosan, T10) in the 2020 season. In contrast, the proportion of Na$^+$ dramatically dropped in the 2021 season from 2.34% (gypsum + PGPR + 0 mg L$^{-1}$ chitosan, T6) to 2.20% (gypsum + PGPR + 25 mg L$^{-1}$ chitosan, T7), 2.04% (gypsum + PGPR + 50 mg L$^{-1}$ chitosan, T8), 2.07% (gypsum + PGPR + 75 mg L$^{-1}$ chitosan, T9), and 2.06% (gypsum + PGPR + 100 mg L$^{-1}$ chitosan, T10) (Table 4).

In the same way, the K$^+$/Na$^+$ ratio increased in the maize treated with gypsum + PGPR + 50 mg L$^{-1}$ chitosan, with rates of 43.5 and 40.4% recorded in the 2020 and 2021 in comparison to the gypsum + PGPR + 0 mg L$^{-1}$ chitosan treatment, respectively (Table 4).

### 3.3. Soil Enzymes Estimations

Data illustrated in Table 5 show that dehydrogenase, urease, amylase, and invertase soil enzyme activities in the rhizosphere of maize plants at two months after sowing were significantly affected by the application of gypsum (without gypsum and with gypsum) and combination of PGPR and various levels of chitosan (0, 25, 50, 75, and 100 mg L$^{-1}$).

The gypsum + PGPR + 50 mg L$^{-1}$ (T8) chitosan treatment resulted in dehydrogenase activity (DHA) measurements of 169.58 and 180.24 mg TPF g$^{-1}$ soil day$^{-1}$ and urease activity measurements of 135.58 and 137.54 NH$_4$$^+$- N g$^{-1}$ soil day$^{-1}$ in the 2020 and 2021 seasons, respectively (Table 5).

The amylase and invertase activity followed the same trend. For instance, during the two growing seasons (2020 and 2021), the highest values for soil amylase activity were recorded under soil amendments with gypsum and grain treated with PGPR, which were 0.360 and 0.370 mg glucose g$^{-1}$ soil h$^{-1}$ for 50 mg L$^{-1}$ chitosan and 0.352 and 0.362 mg glucose g$^{-1}$ soil h$^{-1}$ for 75 mg L$^{-1}$ chitosan, respectively. In the same context, the highest values for soil invertase activity were 0.090 and 0.101 mg sucrose g$^{-1}$ soil h$^{-1}$ for 50 mg L$^{-1}$ chitosan followed by 0.088 and 0.099 mg sucrose g$^{-1}$ soil h$^{-1}$ for 75 mg L$^{-1}$ chitosan during the two growing seasons (2020 and 2021), respectively. In the same context, during the two growth seasons (2020 and 2021), the maximum values for soil invertase activity were 0.090 and 0.101 mg sucrose g$^{-1}$ soil h$^{-1}$ for 50 mg L$^{-1}$ chitosan and 0.088 and 0.099 mg sucrose g$^{-1}$ soil h$^{-1}$ for 75 mg L$^{-1}$ chitosan, respectively, compared to the other studied treatments (Table 5).

**Table 5.** Combined effects of gypsum and bio-priming with PGPR and different levels of chitosan (0, 25, 50, 75, and 100 mg L$^{-1}$), on soil enzyme activities at 60 days from sowing during 2020 and 2021 seasons.

| Treatments | | DHA (mg TPF g$^{-1}$ Soil Day$^{-1}$) | | Urease (NH$_4^+$- N g$^{-1}$ Soil Day$^{-1}$) | | Amylase (mg Glucose g$^{-1}$ soil h$^{-1}$) | | Invertase (mg Sucrose g$^{-1}$ soil h$^{-1}$) | |
|---|---|---|---|---|---|---|---|---|---|
| | | 2020 | 2021 | 2020 | 2021 | 2020 | 2021 | 2020 | 2021 |
| Without Gypsum | T1 | 89.36 ± 0.52 [q] | 101.86 ± 0.54 [q] | 42.36 ± 0.44 [q] | 44.66 ± 0.33 [q] | 0.123 ± 0.003 [o] | 0.131 ± 0.001 [p] | 0.014 ± 0.0004 [p] | 0.016 ± 0.0003 [q] |
| | T2 | 91.91 ± 0.52 [p] | 104.41 ± 0.46 [p] | 44.91 ± 0.41 [p] | 47.21 ± 0.45 [p] | 0.140 ± 0.004 [n] | 0.148 ± 0.003 [o] | 0.016 ± 0.0004 [o] | 0.018 ± 0.0002 [p] |
| | T3 | 100.13 ± 1.45 [n] | 112.63 ± 1.49 [n] | 53.13 ± 0.33 [n] | 55.43 ± 0.36 [n] | 0.193 ± 0.005 [i] | 0.201 ± 0.003 [j] | 0.021 ± 0.0003 [m] | 0.023 ± 0.0004 [n] |
| | T4 | 94.35 ± 0.61 [o] | 106.85 ± 0.51 [o] | 47.35 ± 0.30 [o] | 49.65 ± 0.47 [o] | 0.156 ± 0.003 [l] | 0.164 ± 0.002 [m] | 0.017 ± 0.0004 [n] | 0.019 ± 0.0005 [o] |
| | T5 | 93.44 ± 0.26 [o] | 105.94 ± 1.06 [o] | 46.44 ± 0.22 [o] | 48.74 ± 0.16 [o] | 0.150 ± 0.002 [m] | 0.158 ± 0.003 [n] | 0.017 ± 0.0002 [n] | 0.019 ± 0.0007 [o] |
| | T6 | 101.84 ± 1.69 [m] | 116.95 ± 1.39 [m] | 58.84 ± 0.45 [m] | 62.24 ± 0.19 [m] | 0.149 ± 0.005 [m] | 0.154 ± 0.006 [n] | 0.021 ± 0.0006 [m] | 0.026 ± 0.0004 [m] |
| | T7 | 105.32 ± 1.58 [l] | 120.43 ± 1.08 [l] | 62.32 ± 0.46 [l] | 65.72 ± 0.89 [l] | 0.172 ± 0.004 [k] | 0.177 ± 0.002 [l] | 0.025 ± 0.0005 [l] | 0.030 ± 0.0008 [l] |
| | T8 | 118.12 ± 1.48 [j] | 133.23 ± 1.78 [j] | 75.12 ± 0.41 [j] | 78.52 ± 0.95 [j] | 0.256 ± 0.001 [h] | 0.261 ± 0.003 [i] | 0.037 ± 0.0004 [j] | 0.042 ± 0.0009 [j] |
| | T9 | 107.22 ± 1.40 [k] | 122.33 ± 1.80 [k] | 64.22 ± 0.36 [k] | 67.62 ± 0.59 [k] | 0.184 ± 0.003 [j] | 0.189 ± 0.002 [k] | 0.026 ± 0.0004 [k] | 0.031 ± 0.0003 [k] |
| | T10 | 105.10 ± 1.40 [l] | 122.21 ± 1.11 [k] | 64.10 ± 0.36 [k] | 65.50 ± 0.39 [k] | 0.183 ± 0.003 [j] | 0.188 ± 0.004 [k] | 0.026 ± 0.0004 [k] | 0.032 ± 0.0005 [k] |
| With Gypsum | T1 | 133.17 ± 1.67 [i] | 143.05 ± 1.37 [i] | 94.17 ± 0.86 [i] | 96.05 ± 0.69 [i] | 0.260 ± 0.006 [h] | 0.269 ± 0.005 [h] | 0.052 ± 0.0009 [i] | 0.061 ± 0.0003 [i] |
| | T2 | 134.27 ± 1.67 [h] | 144.15 ± 1.46 [h] | 95.27 ± 0.95 [h] | 97.15 ± 0.86 [h] | 0.268 ± 0.004 [g] | 0.277 ± 0.002 [g] | 0.054 ± 0.0009 [h] | 0.063 ± 0.0004 [h] |
| | T3 | 142.05 ± 2.89 [e] | 151.93 ± 1.99 [e] | 103.05 ± 1.64 [e] | 104.93 ± 0.99 [e] | 0.318 ± 0.006 [c] | 0.327 ± 0.004 [c] | 0.064 ± 0.0012 [e] | 0.073 ± 0.0011 [e] |
| | T4 | 137.57 ± 2.46 [g] | 147.45 ± 1.87 [g] | 98.57 ± 1.68 [g] | 100.45 ± 1.26 [g] | 0.289 ± 0.010 [f] | 0.298 ± 0.009 [f] | 0.058 ± 0.0019 [g] | 0.067 ± 0.0014 [g] |
| | T5 | 139.11 ± 2.55 [f] | 148.99 ± 2.15 [f] | 100.11 ± 1.35 [f] | 101.99 ± 1.25 [f] | 0.299 ± 0.004 [e] | 0.308 ± 0.003 [e] | 0.060 ± 0.0007 [f] | 0.069 ± 0.0003 [f] |
| | T6 | 158.00 ± 2.50 [d] | 168.66 ± 2.90 [d] | 124.00 ± 1.40 [d] | 125.96 ± 1.30 [d] | 0.289 ± 0.003 [f] | 0.299 ± 0.002 [f] | 0.072 ± 0.0008 [d] | 0.083 ± 0.0005 [d] |
| | T7 | 160.92 ± 3.76 [c] | 171.58 ± 3.06 [c] | 126.92 ± 1.66 [c] | 128.88 ± 1.76 [c] | 0.307 ± 0.005 [d] | 0.317 ± 0.007 [d] | 0.077 ± 0.0012 [c] | 0.088 ± 0.0009 [c] |
| | T8 | 169.58 ± 3.80 [a] | 180.24 ± 3.66 [a] | 135.58 ± 1.90 [a] | 137.54 ± 1.40 [a] | 0.360 ± 0.005 [a] | 0.370 ± 0.008 [a] | 0.090 ± 0.0012 [a] | 0.101 ± 0.0018 [a] |
| | T9 | 168.33 ± 3.38 [b] | 178.99 ± 2.74 [b] | 134.33 ± 1.88 [b] | 136.29 ± 1.18 [b] | 0.352 ± 0.002 [b] | 0.362 ± 0.007 [b] | 0.088 ± 0.0006 [b] | 0.099 ± 0.0016 [b] |
| | T10 | 167.92 ± 3.38 [b] | 178.58 ± 2.85 [b] | 133.92 ± 1.98 [b] | 135.88 ± 2.08 [b] | 0.349 ± 0.002 [b] | 0.359 ± 0.010 [b] | 0.087 ± 0.0006 [b] | 0.098 ± 0.0012 [b] |
| **F-test** | | | | | | | | | |
| **Main** | | ** | ** | ** | ** | ** | ** | ** | ** |
| **Sub main** | | ** | ** | ** | ** | ** | ** | ** | ** |
| **Interaction** | | ** | ** | ** | ** | ** | ** | ** | ** |

According to the Duncan's test, means denoted by various letters show significant differences between treatments ($p \leq 0.05$). Values are the means and standard deviations (SD) of three replicates. **DHA**: dehydrogenase; **T1–T10**: as shown in Table 3; **: High significant.

### 3.4. Soil Physical Characteristics

Figure 1 presents the findings of soil bulk density (BD) and total porosity (TP). Both metrics are impacted by various treatments to the same extent but in opposite directions. The effects of gypsum and combination with PGPR, and chitosan on the post-harvest soil BD and TP are minimal.

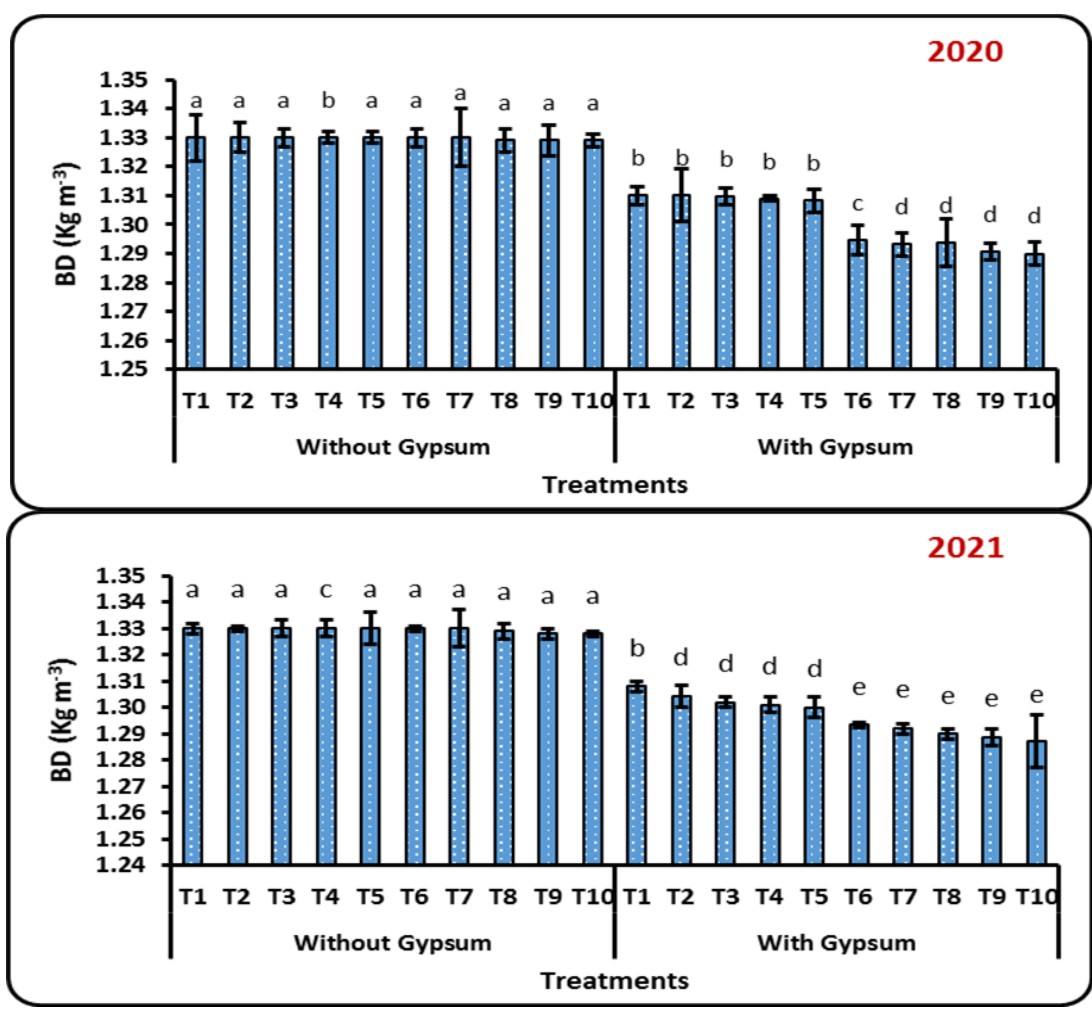

**Figure 1.** *Cont.*

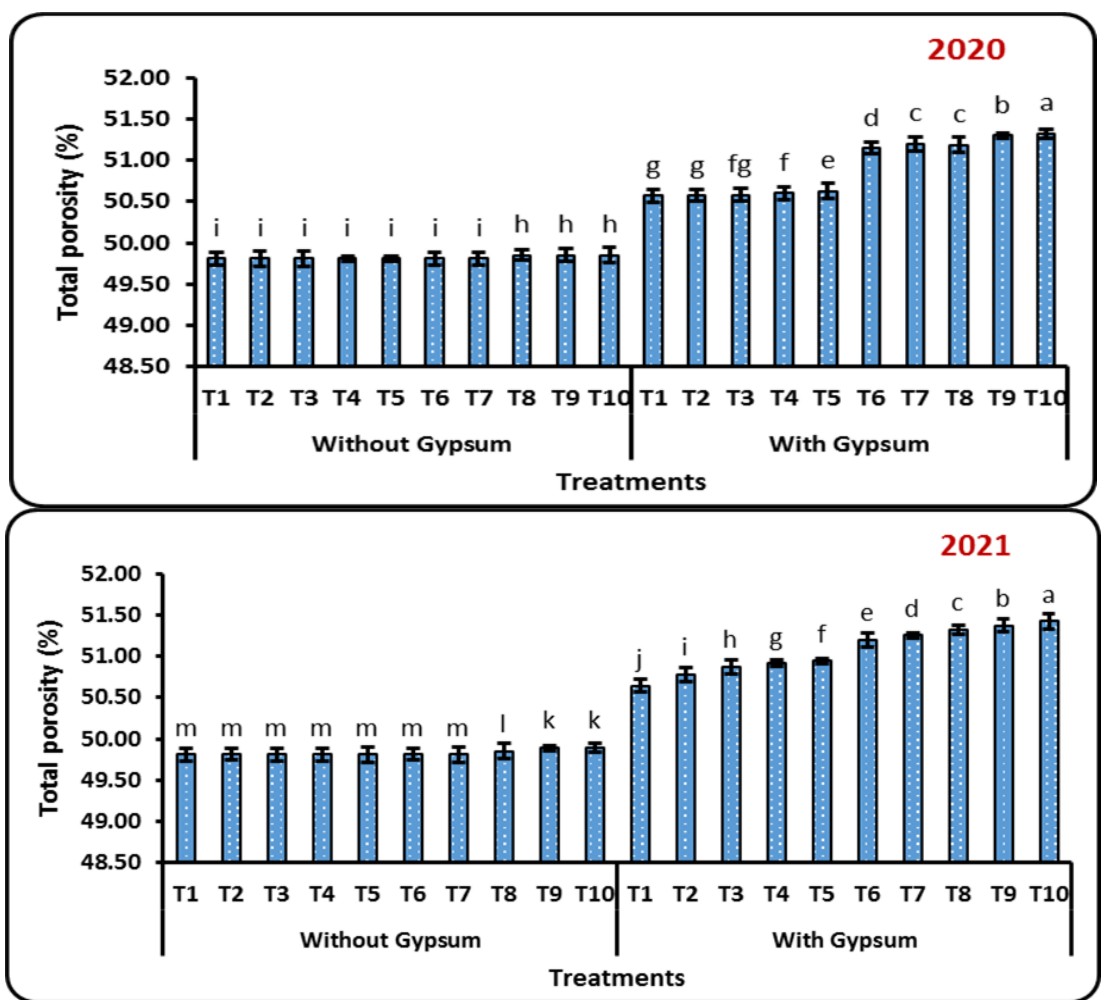

**Figure 1.** Combined effects of gypsum and bio-priming with PGPR and different levels of chitosan (0, 25, 50, 75, and 100 mg L$^{-1}$), on soil physical characteristics at 120 days from sowing during 2020 and 2021 seasons. **BD**: bulk density; **T1–T10**: as shown in Table 3. According to the Duncan's test, means denoted by various letters show significant differences between treatments ($p \leq 0.05$).

Extremely substantial variations were seen for the BD parameters with the application of gypsum + PGPR + 50 mg L$^{-1}$ chitosan treatment (T8), which resulted in 1.29 and 1.29 kg m$^{-3}$ in the 2020 and 2021 seasons, compared to other investigated treatments, respectively. The greatest values for T8 (50 mg L$^{-1}$), T9 (75 mg L$^{-1}$), and T10 (100 mg L$^{-1}$) treatments, on the other hand, were 51.18, 51.30, and 51.32% in season 2020 and 51.32, 51.37, and 51.42% in season 2021, respectively, when maize plants were treated with PGPR and soil was amended with gypsum (Figure 1).

*3.5. Soil Chemical Characteristics*

At four months after sowing (harvest), soil samples from the crop seasons of 2020 and 2021 contained significantly different concentrations ($p \leq 0.05$) of EC (dS m$^{-1}$), ESP (%), and CEC (cmole kg$^{-1}$) based on the applications of gypsum (- gypsum and + gypsum), PGPR (- PGPR and + PGPR) with various concentrations of chitosan (0, 25, 50, 75, and 100 mg L$^{-1}$), (Table 6). The gypsum + PGPR + 50 mg L$^{-1}$ chitosan application (T8) demonstrated reduction values of 5.97 and 4.86 dS m$^{-1}$ for EC and 11.30 and 10.84% for ESP in the 2020 and 2021 seasons, respectively, in comparison to all the tested treatments. The greatest readings, however, were recorded by CEC, which were 37.67 cmole kg$^{-1}$ in 2020 and 39.79 cmole kg$^{-1}$ in the season of 2021 (Table 6).

**Table 6.** Combined effects of gypsum and bio-priming with PGPR and different levels of chitosan (0, 25, 50, 75, and 100 mg L$^{-1}$), on soil chemical characteristics at 120 days from sowing during 2020 and 2021 seasons.

| Treatments | | EC (dS m$^{-1}$) | | ESP (%) | | CEC (Cmol$_e$ kg$^{-1}$) | |
|---|---|---|---|---|---|---|---|
| | | 2020 | 2021 | 2020 | 2021 | 2020 | 2021 |
| **Without Gypsum** | T1 | 8.61 ± 0.28 [a] | 7.26 ± 0.48 [a] | 16.64 ± 0.51 [a] | 16.59 ± 0.66 [a] | 35.97 ± 0.28 [k] | 36.17 ± 0.18 [m] |
| | T2 | 8.60 ± 0.49 [b] | 7.22 ± 0.67 [b] | 16.61 ± 0.72 [b] | 16.55 ± 0.45 [b] | 35.99 ± 0.29 [k] | 36.19 ± 0.17 [lm] |
| | T3 | 8.59 ± 0.39 [b] | 7.22 ± 0.58 [b] | 16.59 ± 0.62 [b] | 16.53 ± 0.46 [bc] | 36.12 ± 0.29 [i] | 36.21 ± 0.18 [kl] |
| | T4 | 6.60 ± 0.41 [b] | 7.20 ± 0.68 [c] | 11.60 ± 0.64 [bc] | 16.52 ± 0.86 [cd] | 36.18 ± 0.23 [ef] | 36.22 ± 0.28 [k] |
| | T5 | 8.59 ± 0.53 [b] | 7.20 ± 0.69 [c] | 16.58 ± 0.76 [c] | 16.50 ± 0.87 [d] | 36.15 ± 0.23 [fgh] | 36.22 ± 0.39 [k] |
| | T6 | 8.57 ± 0.48 [c] | 7.18 ± 0.57 [d] | 16.61 ± 0.71 [bc] | 16.53 ± 0.75 [bc] | 36.17 ± 0.28 [efg] | 36.27 ± 0.37 [j] |
| | T7 | 8.56 ± 0.38 [d] | 7.15 ± 0.59 [e] | 16.58 ± 0.61 [c] | 16.51 ± 0.77 [cd] | 36.18 ± 0.28 [e] | 36.33 ± 0.29 [hi] |
| | T8 | 8.55 ± 0.36 [d] | 7.15 ± 0.49 [e] | 16.57 ± 0.59 [c] | 16.38 ± 0.67 [e] | 36.21 ± 0.26 [d] | 36.53 ± 0.19 [e] |
| | T9 | 8.55 ± 0.58 [d] | 7.14 ± 0.33 [e] | 16.58 ± 0.81 [c] | 16.35 ± 0.51 [f] | 36.23 ± 0.28 [d] | 36.58 ± 0.23 [d] |
| | T10 | 8.55 ± 0.39 [d] | 7.14 ± 0.61 [e] | 16.58 ± 0.62 [c] | 16.36 ± 0.79 [f] | 36.24 ± 0.29 [d] | 36.59 ± 0.15 [d] |
| **With Gypsum** | T1 | 6.63 ± 0.58 [e] | 6.11 ± 0.58 [f] | 11.81 ± 0.81 [d] | 11.55 ± 0.76 [g] | 35.99 ± 0.28 [jk] | 36.31 ± 0.18 [i] |
| | T2 | 6.62 ± 0.47 [ef] | 6.02 ± 0.69 [g] | 11.70 ± 0.70 [e] | 11.40 ± 0.57 [h] | 36.01 ± 0.27 [j] | 36.36 ± 0.29 [h] |
| | T3 | 6.61 ± 0.58 [fg] | 5.97 ± 0.69 [h] | 11.60 ± 0.81 [f] | 11.34 ± 0.87 [i] | 36.15 ± 0.28 [gh] | 36.44 ± 0.39 [g] |
| | T4 | 6.60 ± 0.68 [g] | 5.96 ± 0.71 [hi] | 11.60 ± 0.91 [f] | 11.31 ± 0.89 [j] | 36.18 ± 0.28 [e] | 36.47 ± 0.40 [f] |
| | T5 | 6.60 ± 0.49 [g] | 5.96 ± 0.53 [i] | 11.59 ± 0.72 [f] | 11.25 ± 0.71 [k] | 36.17 ± 0.29 [ef] | 36.49 ± 0.23 [f] |
| | T6 | 6.07 ± 0.27 [h] | 5.02 ± 0.58 [j] | 11.43 ± 0.50 [g] | 11.30 ± 0.56 [j] | 37.57 ± 0.27 [c] | 39.50 ± 0.18 [c] |
| | T7 | 6.00 ± 0.49 [i] | 4.98 ± 0.61 [k] | 11.38 ± 0.72 [h] | 11.18 ± 0.79 [l] | 37.61 ± 0.29 [b] | 39.58 ± 0.33 [b] |
| | T8 | 5.97 ± 0.39 [j] | 4.86 ± 0.66 [l] | 11.30 ± 0.62 [i] | 10.84 ± 0.84 [m] | 37.67 ± 0.29 [a] | 39.79 ± 0.36 [a] |
| | T9 | 5.96 ± 0.33 [j] | 4.86 ± 0.38 [l] | 11.28 ± 0.56 [i] | 10.81 ± 0.56 [n] | 37.68 ± 0.23 [a] | 39.80 ± 0.28 [a] |
| | T10 | 5.97 ± 0.61 [j] | 4.86 ± 0.59 [l] | 11.28 ± 0.54 [i] | 10.81 ± 0.77 [n] | 37.69 ± 0.26 [a] | 39.79 ± 0.39 [a] |
| **F-test** | | | | | | | |
| **Main** | | ** | ** | ** | ** | ** | ** |
| **Sub main** | | ** | ** | ** | ** | ** | ** |
| **Interaction** | | ** | ** | ** | ** | ** | ** |

According to the Duncan's test, means denoted by various letters show significant differences between treatments ($p \leq 0.05$). Values are the means and standard deviations (SD) of three replicates. EC: electrical conductivity; ESP: exchangeable sodium percentage; CEC: cation exchange capacity; **T1–T10**: as shown in Table 3; **: High significant.

According to the results, the gypsum + PGPR + 50 mg L$^{-1}$ chitosan treatment (T8) was the most effective at reducing EC and ESP and increasing CEC in salt-affected soils when compared to other investigated treatments.

### 3.6. Yield of Maize and Its Components

The maize yield and its components, including ear length (cm), ear diameter (cm), 100 grain weight (g), and grain yield (Kg ha$^{-1}$), were considerably reduced on salt-affected soils throughout the 2020 and 2021 seasons in the absence of gypsum, PGPR, and chitosan treatments (Table 7). However, when maize plants were treated with gypsum and combinations of PGPR with chitosan, these adverse effects were dramatically reduced.

**Table 7.** Combined effects of gypsum and bio-priming with PGPR and different levels of chitosan (0, 25, 50, 75, and 100 mg $L^{-1}$), on yields and yield components of maize plants during 2020 and 2021 seasons.

| Treatments | | Ear Length (cm) | | Ear Diameter (cm) | | 100-Grain Weight (g) | | Grain Yield (kg ha$^{-1}$) | |
|---|---|---|---|---|---|---|---|---|---|
| | | 2020 | 2021 | 2020 | 2021 | 2020 | 2021 | 2020 | 2021 |
| Without Gypsum | T1 | 16.82 ± 1.78 [o] | 16.91 ± 1.10 [o] | 4.21 ± 0.52 [n] | 4.24 ± 0.22 [n] | 31.54 ± 1.18 [n] | 32.94 ± 1.11 [n] | 5361.80 ± 39.16 [n] | 5376.00 ± 91.49 [o] |
| | T2 | 17.30 ± 1.10 [n] | 17.39 ± 1.12 [n] | 4.33 ± 0.32 [m] | 4.36 ± 0.19 [m] | 32.44 ± 3.18 [m] | 33.84 ± 1.14 [m] | 5514.80 ± 61.46 [m] | 5529.10 ± 71.16 [n] |
| | T3 | 18.85 ± 2.08 [j] | 18.94 ± 1.78 [j] | 4.71 ± 0.22 [h] | 4.74 ± 0.34 [h] | 35.34 ± 1.16 [h] | 36.74 ± 2.35 [i] | 6007.80 ± 66.99 [i] | 6022.50 ± 76.57 [j] |
| | T4 | 17.76 ± 1.10 [l] | 17.85 ± 1.70 [l] | 4.44 ± 0.62 [k] | 4.47 ± 0.29 [k] | 33.30 ± 1.18 [k] | 34.70 ± 1.29 [k] | 5661.00 ± 70.60 [k] | 5676.50 ± 53.39 [l] |
| | T5 | 17.59 ± 2.05 [m] | 17.68 ± 1.95 [m] | 4.40 ± 0.81 [l] | 4.43 ± 0.46 [l] | 32.98 ± 1.09 [l] | 34.38 ± 2.48 [l] | 5606.60 ± 55.58 [l] | 5621.20 ± 64.60 [m] |
| | T6 | 18.22 ± 2.12 [k] | 18.33 ± 1.62 [k] | 4.34 ± 0.93 [m] | 4.38 ± 0.59 [m] | 33.23 ± 2.22 [k] | 34.89 ± 2.95 [k] | 5815.60 ± 79.12 [j] | 5840.40 ± 69.10 [k] |
| | T7 | 18.85 ± 2.10 [j] | 18.96 ± 2.00 [j] | 4.49 ± 0.72 [j] | 4.53 ± 0.52 [j] | 34.37 ± 3.19 [j] | 36.03 ± 2.49 [j] | 6014.52 ± 53.15 [i] | 6039.82 ± 63.59 [j] |
| | T8 | 21.14 ± 1.09 [h] | 21.25 ± 1.39 [h] | 5.03 ± 0.12 [de] | 5.07 ± 0.79 [de] | 38.54 ± 3.16 [e] | 40.20 ± 2.48 [f] | 6745.08 ± 47.31 [f] | 6770.28 ± 87.34 [g] |
| | T9 | 19.19 ± 2.07 [i] | 19.30 ± 2.00 [i] | 4.57 ± 0.72 [i] | 4.61 ± 0.87 [i] | 34.99 ± 3.13 [i] | 36.65 ± 1.59 [i] | 6123.02 ± 62.59 [h] | 6148.32 ± 92.69 [i] |
| | T10 | 19.16 ± 1.07 [i] | 19.27 ± 2.04 [i] | 4.56 ± 0.32 [i] | 4.60 ± 0.28 [i] | 34.95 ± 3.13 [i] | 36.61 ± 2.96 [i] | 6115.78 ± 72.59 [h] | 6140.28 ± 62.78 [i] |
| With Gypsum | T1 | 21.79 ± 2.11 [g] | 21.92 ± 1.91 [g] | 4.95 ± 0.62 [f] | 5.00 ± 0.49 [f] | 39.79 ± 2.16 [c] | 41.57 ± 1.91 [c] | 7161.60 ± 58.99 [c] | 7190.10 ± 86.67 [c] |
| | T2 | 21.97 ± 1.11 [f] | 22.10 ± 1.18 [f] | 4.99 ± 0.52 [e] | 5.04 ± 0.47 [e] | 36.12 ± 3.13 [g] | 37.90 ± 1.39 [h] | 6501.12 ± 73.98 [g] | 6530.62 ± 29.91 [h] |
| | T3 | 23.24 ± 3.15 [c] | 23.37 ± 2.11 [c] | 5.28 ± 0.83 [a] | 5.33 ± 0.36 [a] | 36.07 ± 3.13 [g] | 37.85 ± 1.49 [h] | 6493.44 ± 53.98 [g] | 6522.74 ± 56.98 [h] |
| | T4 | 22.51 ± 2.24 [e] | 22.64 ± 1.04 [e] | 5.12 ± 0.85 [c] | 5.17 ± 0.27 [c] | 38.74 ± 3.20 [e] | 40.52 ± 1.69 [e] | 6973.44 ± 35.19 [e] | 7002.94 ± 59.09 [f] |
| | T5 | 22.76 ± 2.09 [d] | 22.89 ± 2.27 [d] | 5.17 ± 0.92 [b] | 5.22 ± 0.49 [b] | 39.06 ± 2.20 [d] | 40.84 ± 2.97 [d] | 7031.04 ± 65.19 [d] | 7060.34 ± 78.69 [e] |
| | T6 | 22.12 ± 2.07 [f] | 22.26 ± 1.47 [f] | 4.71 ± 0.31 [h] | 4.77 ± 0.64 [h] | 37.92 ± 3.12 [f] | 39.72 ± 3.59 [g] | 6939.36 ± 41.96 [e] | 6973.76 ± 77.86 [f] |
| | T7 | 22.53 ± 3.11 [e] | 22.67 ± 2.81 [e] | 4.79 ± 0.52 [g] | 4.85 ± 0.78 [g] | 38.62 ± 3.18 [e] | 40.42 ± 2.49 [e] | 7067.46 ± 33.54 [d] | 7101.96 ± 89.74 [d] |
| | T8 | 23.74 ± 3.11 [a] | 23.88 ± 2.91 [a] | 5.05 ± 0.72 [d] | 5.11 ± 0.97 [d] | 40.70 ± 4.19 [a] | 42.50 ± 3.98 [a] | 7448.10 ± 35.30 [a] | 7482.30 ± 93.80 [a] |
| | T9 | 23.57 ± 3.05 [b] | 23.71 ± 2.25 [b] | 5.01 ± 0.71 [de] | 5.07 ± 0.59 [de] | 40.40 ± 3.09 [b] | 42.20 ± 3.93 [b] | 7393.20 ± 56.77 [b] | 7427.60 ± 69.07 [b] |
| | T10 | 23.51 ± 2.05 [b] | 23.65 ± 1.85 [b] | 5.00 ± 0.21 [e] | 5.06 ± 0.87 [e] | 40.30 ± 4.29 [b] | 42.10 ± 2.92 [b] | 7374.90 ± 86.77 [b] | 7408.50 ± 47.17 [b] |
| **F-test** | | | | | | | | | |
| **Main** | | ** | ** | ** | ** | ** | ** | ** | ** |
| **Sub main** | | ** | ** | ** | ** | ** | ** | ** | ** |
| **Interaction** | | ** | ** | ** | ** | ** | ** | ** | ** |

According to the Duncan's test, means denoted by various letters show significant differences between treatments ($p \leq 0.05$). Values are the means and standard deviations (SD) of three replicates; **T1–T10**: as shown in Table 3; **: High significant.

In comparison to all the tested treatments, plants treated with gypsum + PGPR + 50 mg L$^{-1}$ chitosan (T8) during the 2020 and 2021 growing seasons recorded the highest ear length, ear diameter, 100 grain weight, and grain yield, measuring 23.74 and 23.88 cm, 5.05 and 5.11 cm, and 40.70 and 42.50 g, 7448.10 and 7482.30 Kg ha$^{-1}$, respectively (Table 7). As a consequence, the yield of maize and its constituent parts occurred at different chitosan concentrations under gypsum + PGPR treatments in the declining order of 50 > 75 > 100 > 25 > 0 mg L$^{-1}$.

These findings showed that the negative effects of salt-affected soils on maize plants might be significantly lessened by the application of gypsum + PGPR and chitosan.

## 4. Discussion

The type of salinity and the accessibility of soil amendments that could mitigate the effects of salinity on soils determine how salt-affected soils can be improved. Among the soil inputs that have been consistently acknowledged to enhance the biological, physical, and chemical features of salty soils for higher food production are gypsum, PGPR and chitosan amendments. These amendments refer to the coordinated use of beneficial microorganisms and organic nutrient sources during crop cultivation to boost output. Increased soil organic matter, necessary nutrient and water availability, soil structure, and microbial activity are all factors that bio-organic additions can use to promote crop and soil production.

### 4.1. Photosynthetic Pigments

Due to the fact that fewer photosynthetic pigments are destroyed, salt stress may have a negative effect on photosynthesis. Additionally, these diminished effects might be caused by a rise in the manufacture of proteolytic enzymes like chlorophyllase, which is the primary culprit in the destruction of chlorophyll and/or the damage of the photosynthetic system [42]. Additionally, these are crucial regulatory actions to reduce high light absorption and hence lower ROS production [43]. Lowered pigment biosynthesis and oxidation of photosynthetic pigments caused damage to the photosynthetic apparatus, which reduced the effect of stress and decreased photosynthetic carbon uptake [44]. Depending on the applications of the gypsum + PGPR + 50 mg L$^{-1}$ chitosan treatment, maize leaves from the 2020 and 2021 cropping seasons contained significantly different levels ($p < 0.05$) of chlorophyll, carotenoids, and proline at 60 days after sowing (Table 3).

The increased availability of amino compounds released from chitosan or an increase in cytokinin content and plant water balance that encouraged chlorophyll production may have contributed to these enhanced findings [45]. These outcomes are consistent with those of [46], who discovered that *Phaseolus vulgaris* photosynthetic pigments were enhanced following chitosan treatment. According to [30], chitosan treatments may increase the effects of salt stress reduction and promote sunflower plant growth and photosynthetic pigments. Additionally, [45], the application of chitosan and PGPR (*Bacillus thuringiensis*) improves the chlorophyll pigments of sweet pepper while reducing the negative effects of salinity.

### 4.2. N, K$^+$, Na$^+$ and the K$^+$/Na$^+$ Ratio

Our findings indicated that the application of gypsum, PGPR, and chitosan considerably enhanced maize development, which can reduce salt stress by adsorbing Na$^+$ ions and releasing non-toxic, advantageous N and K$^+$ ions that have a positive impact on the soil and plant health compared to untreated soil (Table 4). On the other hand, through mineralization and immobilization, soil microorganisms play a crucial role in nutrient cycling and as a result, have a favorable impact on nutrient availability and organic matter.

According to [47], which supported our findings, adding soil ameliorants (gypsum and sulfur) along with the proper PGPR strains is a crucial technique to improve N uptake in cowpea plants growing in salty soils. Similar to this, the exopolysaccharide produced by PGPR, which binds the Na$^+$ ions in the soil and reduces their absorption, may be the cause of the reduced concentration of Na$^+$ ions in the leaves of maize plants treated with PGPR

(*A. lipoferum* + *B. circulance*) under saline soil [48]. In addition, in line with our findings, the application of chitosan boosted the concentration of $K^+$ in the shoots of the maize plant while simultaneously lowering the concentration of $Na^+$ [49].

### 4.3. Soil Enzymes Estimations

Dehydrogenase, urease, amylase, and invertase were chosen as the variations in soil enzyme activities because of the variety of biological functions they provide in the soil. Dehydrogenase was measured to assess microbial activity in soil, and urease was measured to understand the function of microorganisms in the nitrogen cycle in soil. Additionally, the choice of amylase and invertase was influenced by the importance of these enzymes to the soil's carbon cycle. Compared to the other investigated treatments, the gypsum + PGPR + 50 mg $L^{-1}$ chitosan treatment increased soil enzyme activity in the rhizosphere of maize plants during the two growing seasons 2020 and 2021 (Table 5).

Gypsum, PGPR, and chitosan application increased soil enzyme activity under salt stress in the current study, which may hasten aerobic organisms' metabolic activities that are crucial for regulating the release of bioavailable nutrients from organic molecules [50,51]. The primary explanation is that soil enzyme activity is closely correlated with the types, numbers, and abundance of soil microbes [52], and the metabolism and reproduction of a soil microbial community have been adopted to be influenced by temperature and precipitation [53]. These findings are confirmed in rice plants [54,55]; Eucalyptus plants [56]; maize plants [48]; wheat plants [57].

### 4.4. Soil Physicochemical Characteristics

Chitosan and beneficial bacteria have a significant impact on soil fertility enhancement; therefore, combining them with the application of gypsum has the potential to significantly improve saline soils (Figure 1 and Table 6). The integrated use between gypsum and bio-organic amendments in the reclamation of salt-affected soils has the potential to be important, but only a few normative studies have been carried out in this area at the field level. Gypsum's solubility can be increased for greater efficacy in treating saline soils by inoculating it with beneficial microbes prior to applying it [58]. Additionally, it has been demonstrated that applying gypsum in combination with beneficial microbes might lessen the impact of soil salinity on wheat yield by decreasing the SAR, EC, and ESP [59]. Similarly, gypsum was used to improve a saline–sodic soil along with two bacterial species (*B. megaterium* and *B. subtilis*) and two species of fungal (*Alternaria* spp. and *Aspergillus* spp.). This improved the soil's saturated hydraulic conductivity, which facilitates simple water movement for plant use [58]. Similar to this, the effect of gypsum addition with PGPR on soil bulk density and porosity may be to decrease salinity and lowering exchangeable $Na^+$, which decreases soil dispersion and increases soil porosity, soil aggregation, and causes a net reduction in bulk density [60,61].

### 4.5. Yield of Maize and Its Components

In comparison to the other investigated treatments, yield and yield-components of maize plants growing in salty soils increased by treatment with gypsum, inoculated with *A. lipoferum* + *B. subtilis*, and 50 mg $L^{-1}$ chitosan (Table 7). In addition, the formation of growth regulators such as gibberellins, auxins, and cytokinins, increase in proline content, and the enhancement of the different properties of the soil had positive effects on the yield of maize, which ultimately resulted in increased crop production [48].

These results are corroborated by earlier research; Al Kahtani [45], demonstrated that treating seeds with *B. thuringiensis* and chitosan was an efficient and less expensive method to deal with the negative effects of salinity on fruit yield characteristics of sweet pepper. Additionally, the yield of wheat and maize plants increased when treated with PGPR (*A. lipoferum* and *Enterobacter cloacae*), gypsum, and when not irrigated with high-quality water [62]. According to [48], under salt-affected soil conditions, inoculation treatments

with PGPR combined with phosphogypsum had a substantial impact on grain yield and yield-related characteristics during 2019–2020 seasons.

## 5. Conclusions

The findings of the current study demonstrate that, in saline soil conditions, gypsum, bacterial inoculation, and chitosan can be used to promote the growth and nutrient intake of maize plants. With an increase in yield, this method is seen to be a viable one for reducing the negative effects of saline–sodic soil on plant growth. Accordingly, the findings imply that applying a gypsum + PGPR + 50 mg $L^{-1}$ chitosan treatment can considerably improve plant physiology, nutrient absorption, soil enzyme activity, soil physicochemical parameters, and yield in maize plants.

**Author Contributions:** Conceptualization, A.E.-D.O., M.M.A., M.E. and S.E.-N.; methodology, A.E.-D.O., M.M.A., M.E. and S.E.-N.; software, A.E.-D.O. and M.M.A.; validation, A.E.-D.O. and S.E.-N.; formal analysis, A.E.-D.O., M.M.A. and S.E.-N.; investigation, A.E.-D.O., M.M.A. and M.E.; resources, A.E.-D.O., M.M.A. and S.E.-N.; data curation, A.E.-D.O., M.M.A. and S.E.-N.; writing—original draft preparation, A.E.-D.O.; writing—review and editing, A.E.-D.O.; visualization, A.E.-D.O., M.M.A. and S.E.-N.; supervision, A.E.-D.O.; funding acquisition, M.E. All authors have read and agreed to the published version of the manuscript.

**Funding:** This research received no external funding.

**Institutional Review Board Statement:** Not applicable.

**Informed Consent Statement:** Not applicable.

**Data Availability Statement:** The data are available upon request.

**Acknowledgments:** The authors extend their appreciation to the Deanship of Scientific Research at King Khalid University for funding this work through the Large Groups Project under grant number L.G.P. 2/138/43. All of the authors are grateful for the support provided by the Soils, Water, and Environment Research Institute (SWERI), Agriculture Research Center (ARC), Egypt.

**Conflicts of Interest:** The authors declare no conflict of interest.

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
