# Peer review of "Impact of Gypsum and Bio-Priming of Maize Grains on Soil Properties, Physiological Attributes and Yield under Saline–Sodic Soil Conditions"

_agronomy, doi:10.3390/agronomy12102550_

Round 1
Reviewer 1 Report
dear authors
You need to spend effort and time to reduce the plagiarism percentage in your manuscript. The current percentage is about 26% which is not accepted by highly impacted journals
with regards

Author Response
First, thank you very much for the message to let us to that our paper can be accepted after corrections. We heartily appreciate the effort by the reviewer and the editor and think that the comments are of great value to improve the quality of our paper.
We decrease the similarity of our manuscript

Reviewer 2 Report
The manuscript deals with a field experiment to evaluate gypsum, PGPR and chitosan for the improvement of maize production in Egypt. The manuscript needs improvement to help the reader identify the most relevant results of the experiment and to increase scientific rigor.
General comments:
- In the introduction, some information/data related with maize cultivation in the area and why the results of the experiment are relevant to answer area-specific questions (e.g water productivity, yield gap) should be added
- The Material and Methods section needs to be improved. Please start from the description of the location where the field experiment was carried out and see “specific comments”
- Results: the experiment provides a very large amount of results. Their visualization should be simplified.
Specific comments:
- Lines 31-36: please verify data on the global problem soil salinity and also provide regional-scale data. For instance I have found the following for the global-scale: Corwin, D. L. (2021). Climate change impacts on soil salinity in agricultural areas. European Journal of Soil Science, 72(2), 842-862; Shahid, S. A., Zaman, M., & Heng, L. (2018). Soil salinity: Historical perspectives and a world overview of the problem. In Guideline for salinity assessment, mitigation and adaptation using nuclear and related techniques (pp. 43-53). Springer, Cham.
- Line36: reformulate the sentence. For instance, substituting “additionally” with “the reason of crop productivity reduction is that
- Lines 82-87: The actual amount of Gypsum applied seems not to be reported
- Lines 105-108: the arrangement of plots is not clear, in my opinion (is 60 cm the interrow distance? Was the sowing done manually? How was the distance between treatment (100cm) managed?
- Line 112: reported concentration of PGPR is not clear to me
- Line 113: which type of “amendments”? if only chemical fertilizers were supplied eliminate “in addition to applying soil amendment”
- Line 188-189: please specify Factors and interactions considered and if Year is included in the ANOVA or the analyses were carried out separately for each year.
- Line 188-189: provide the reference to the software utilized
- Figure 1, 2 and 3: it is difficult to identify the treatments and the significant differences among treatments. Please consider to visualize in a different way, especially if year were analyzed separately
Round 2
Reviewer 1 Report
Dear Authors
Thanks a lot for the time and effort that you spent to improve your manuscript. However, I still see your MS needs additional time and effort to be publishable in Agronomy, MDPI. Please consider the following comments and observations:
1- Your experimental design was not appropriate for your hypothesis, you conducted the experiments following split-split plot design considering that you have 3 different factors, Gypsum (with and without), PGPR (with and without), and Chitosan (0, 25, 50, 75 and 100g). But you in reality have only two factors : Gypsum (with and without) and grain priming (10 treatments including control treatment).
2- Based on the above mentioned comment, you will be required to do the following:
a- reanalyses the data again
b- rewrite the results
c- rewrite the discussion
d- rewrite the Abstract and conclusion
3- The title will be changed based on the new phenomena
4- you must improve your introduction to meet the new phenomena and add a section about seeds/grain priming to alleviate salinity tolerance in maize/cereals
5- Add new references about seeds/grain priming to enhance salinity tolerance
6- It will be a great support for your work to add a supplementary file for ANOVA tables, few photos of your field experiment and any other beneficial materials
best wishes
magdi
Reviewer 2 Report
The authors have answered to the comments received, but one point still needs to be resolved. The display of results is still not clear in Fig. 1-3 (comment item 9). I suggest reporting the order of treatments as in the tables, showing the same labels along the abscissa instead of T1-T20
Round 3
Reviewer 1 Report
Dear Authors
Thanks a lot for the time and effort that you spent to improve your manuscript. However, I still see your MS needs additional time and effort to be publishable in Agronomy, MDPI. Please consider the following comments and observations:
1- Your experimental design was not appropriate for your hypothesis, you conducted the experiments following split-split plot design considering that you have 3 different factors, Gypsum (with and without), PGPR (with and without), and Chitosan (0, 25, 50, 75 and 100g). But you in reality have only two factors : Gypsum (with and without) and grain priming (10 treatments including control treatment).
2- Based on the above mentioned comment, you will be required to do the following:
a- reanalyses the data again
b- rewrite the results
c- rewrite the discussion
d- rewrite the Abstract and conclusion
3- The title will be changed based on the new phenomena
4- you must improve your introduction to meet the new phenomena and add a section about seeds/grain priming to alleviate salinity tolerance in maize/cereals
5- Add new references about seeds/grain priming to enhance salinity tolerance
6- It will be a great support for your work to add a supplementary file for ANOVA tables, few photos of your field experiment and any other beneficial materials
best wishes
magd
